# An Evaluation of the Mechanical Properties of a Hybrid Composite Containing Hydroxyapatite

**DOI:** 10.3390/ma16134548

**Published:** 2023-06-23

**Authors:** Leszek Klimek, Karolina Kopacz, Beata Śmielak, Zofia Kula

**Affiliations:** 1Institute of Materials Science and Engineering, Faculty of Mechanical Engineering, Lodz University of Technology, ul. B. Stefanowskiego 1/15, 90-924 Lodz, Poland; leszek.klimek@p.lodz.pl; 2“Dynamo Lab” Academic Laboratory of Movement and Human Physical Performance, Medical University of Lodz, ul. Pomorska 251, 92-213 Lodz, Poland; karolina.kopacz@umed.lodz.pl; 3Department of Dental Prosthodontics, Medical University of Lodz, ul. Pomorska 251, 92-213 Lodz, Poland; 4Department of Dental Technology, Medical University of Lodz, ul. Pomorska 251, 92-213 Lodz, Poland; zofia.kula@umed.lodz.pl

**Keywords:** composite mechanical properties, composite tribological properties, dental composites, hydroxyapatite

## Abstract

There is currently a lack of scientific reports on the use of composites based on UDMA resin containing HAp in conservative dentistry. The aim of this study was therefore to determine the effect of hydroxyapatite content on the properties of a hybrid composite used in conservative dentistry. This paper compares a commercial hybrid composite with experimental composites treated with 2% by weight (b/w), 5% b/w, and 8% b/w hydroxyapatite. The composites were subjected to bending strength, compression, and diametrical compression tests, as well as those for impact strength, hardness, and tribological wear. The obtained results were subjected to statistical analysis. Increased hydroxyapatite was found to weaken the mechanical properties; however, 2% b/w and 5% b/w hydroxyapatite powder was found to achieve acceptable results. The statistical analysis showed no significant differences. HAp is an effective treatment for composites when applied at a low concentration. Further research is needed to identify an appropriate size of HAp particles that can be introduced into a composite to adequately activate the surface and modification its composition.

## 1. Introduction

In dental practice, the ideal material for the reconstruction of hard tooth tissues is still being sought. The most used are composite materials based on polymers [1]. These materials consist of a polymer matrix (UDMA, Bis-GMA TEGDMA), a binding agent (vinyl silane), compounds regulating the polymerization process (initiators, inhibitors), substances conditioning aesthetic effects (dyes, UV absorbers and others), and filler particles (silicon oxide, quartz, colloidal silica, boron glass, alumina-lithium) [2,3,4,5]. The filler is one of the basic components of composites which constitutes 35 to 70% of the mass of the material [6]. Fillers are added to improve mechanical properties such as bending strength, fracture toughness, and abrasion resistance [7,8]. Fillers can be divided into macro-fillers and micro-fillers based on the size of the filler particles in the polymer matrix. There are also hybrid composites which contain different sizes of filler particles; these are characterized by favorable mechanical properties [9]. Indeed, the best properties are characterized by hybrid materials with filler particles below 0.1 µm (microhybrid and nanohybrid) [7,8,9]. Hydroxyapatite (HAp) has certain biological properties that make it a suitable filler. HAp, i.e., Ca_3_(PO_4_)_2_, is part of the hard tissues of teeth and bones [10,11]. It is characterized by high biocompatibility and bioactivity [12,13,14,15]. It does not cause inflammation; it is not an irritant nor does it demonstrate toxic or carcinogenic effects [16,17,18]. Hydroxyapatite is used to fill bone defects, coat implants, and as an active agent against tooth hypersensitivity in toothpastes [10,11,12,19]. The addition of hydroxyapatite as a filler affects the mechanical properties of the dental filling [19,20,21]. Depending on the amount and form of the introduced filler, it may improve or deteriorate the mechanical properties. Santos et al. [19] showed an improvement in strength properties after adding 3% hydroxyapatite in the form of nanofibers to a TEGDMA/Bis-GMA polymer matrix. Domingo et al. [20] found a 30% improvement in mechanical properties after adding HAp in the form of powder. In turn, Elkassas and Arafa [21] and Priyadarsini et al. [22] confirmed that the addition of HAp on the nanometric scale improved to better understand the properties of composite materials containing hydroxyapatite fillers, and studies have examined the effect of hydroxyapatite treatment on selected mechanical properties [23,24,25]. One such property is hardness, which is easy to measure and comparable with other findings. It determines the ability of the material to resist deformation. However, to fully characterize the material, it is necessary to analyze the loads it will be subjected to and perform appropriate strength tests. During the chewing process, the teeth are exposed to various types of mechanical loads (bending, compression). Sometimes, the loads may have a shock character. The basic strength test is the static tensile test, which provides an insight into a number of important parameters characterizing both strength and plastic properties, as well as material constants. It is often used in studies on the mechanisms of deformation and fracture of materials. It is not generally used for materials used for dental fillings because it requires relatively large samples, and hence has a higher cost, and also entails various technical problems. Therefore, it is typically replaced by a diametrical compression test. 

While existing studies on doping with hydroxyapatite have been based on composites based on TEGDMA, Bis-GMA, HEMA resins, and resin-modified glass ionomer cements [23,24,25,26,27,28,29], the present paper examines the hydroxyapatite modification of a UDMA resin dental composite, which is used in restorative dentistry. This type of composite enables direct reconstruction of all cavities according to Black’s classification. Furthermore, the present work uses micro-scale hydroxyapatite, while previous studies have examined nano-sized hydroxyapatite [28,29,30], using hydroxyapatite in powder form. In addition to the typical mechanical loads, teeth are subject to wear processes, e.g., abrasion, when chewing and grinding food. Wear resistance determines the service life of the dental filling. 

The aim of this paper is to study compare the hardness, static strength properties (compression and bending), dynamic properties (impact strength and fracture toughness), and wear resistance of three experimental composites with different HAp contents with those of a commercial filler. The null hypothesis assumes that the use of hydroxyapatite as a filler in dental composites affects its mechanical properties.

## 2. Materials and Methods

Samples were prepared for testing as rectangular (*n* = 120) and cylindrical (*n* = 120) beams of appropriate sizes in accordance with ISO standards [20,21,22,31]. The samples consisted of a commercial composite material based on urethane dimethacrylate (UDMA) (Gradia Direct, GC Tokyo, Japan) (*n* = 60) as a reference, and three test substances with self-sintered hydroxyapatite (HAp): 2% b/w HAp (administered as 30 µm grain size) (*n* = 60), 5% b/w (*n* = 60) and 8% b/w (*n* = 60). Experimental material was prepared in accordance with ISO standards [31,32,33,34,35]. Table 1 shows the content and size of the filler in individual samples.

Hydroxyapatite was synthesized by the wet method. The dried HAp grains obtained were fractionated using an LPzE-3e laboratory shaker (MULTISERW-Morek, Brzeźnica, Poland) and passed through a set of three sieves: 0.1 mm, 0.05 mm, and 0.025 mm. The filler was then introduced into the composite material using a Roti-Speed stirrer (Carl Roth GmbH + Co., KG, Karlsruhe, Germany). This stirrer is used to mix very small samples in micro tubes at 5000 rpm. for about 5 min. The process of mixing was carried out in a darkened room under standardized conditions of temperature and humidity. The resulting material was stored in polypropylene syringes with a plunger. The samples were prepared for the strength by placing the composite in a silicone mold between basic laboratory slides to protect the surface against oxygen inhibition. Then, each layer of material, with a thickness of 1 mm, was irradiated for 20 s using a diode polymerization lamp (Elipar S10, 3M ESPE, St. Paul, MS, USA) with a real power of 1400 mW/cm^2^, emitting light radiation in the range of 450–490 nm. Before the mechanical tests, the samples were artificially aged by incubation at 37 °C in distilled water for 24 h. Then, tests of bending strength, compression, diametrical compression, impact tests, hardness measurements, and tribological wear resistance tests were carried out (Table 2).

### 2.1. Bending Strength Test

Ten rectangular beam-shaped samples (2 mm × 2 mm × 25 mm) from each series were prepared for the three-point bending strength test. The tests were performed with a UMT TriboLab Bruker multifunctional device (Bruker, Karlsruhe, Germany). The travel speed of the traverse was 0.5 mm/min while maintaining the support spacing of 20 mm. The radii of the supports and the mandrel implementing the excitation were 1 mm. Figure 1 shows a photo of a sample placed in the device during the test. For comparative purposes, the strength was calculated according to the following formula:δ = 3FL/2bh^2^
where

δ—flexural strength [MPa];

F—destructive force [N];

L—spacing of supports [mm];

b—sample width [mm];

h—sample thickness [mm].

**Figure 1 materials-16-04548-f001:**
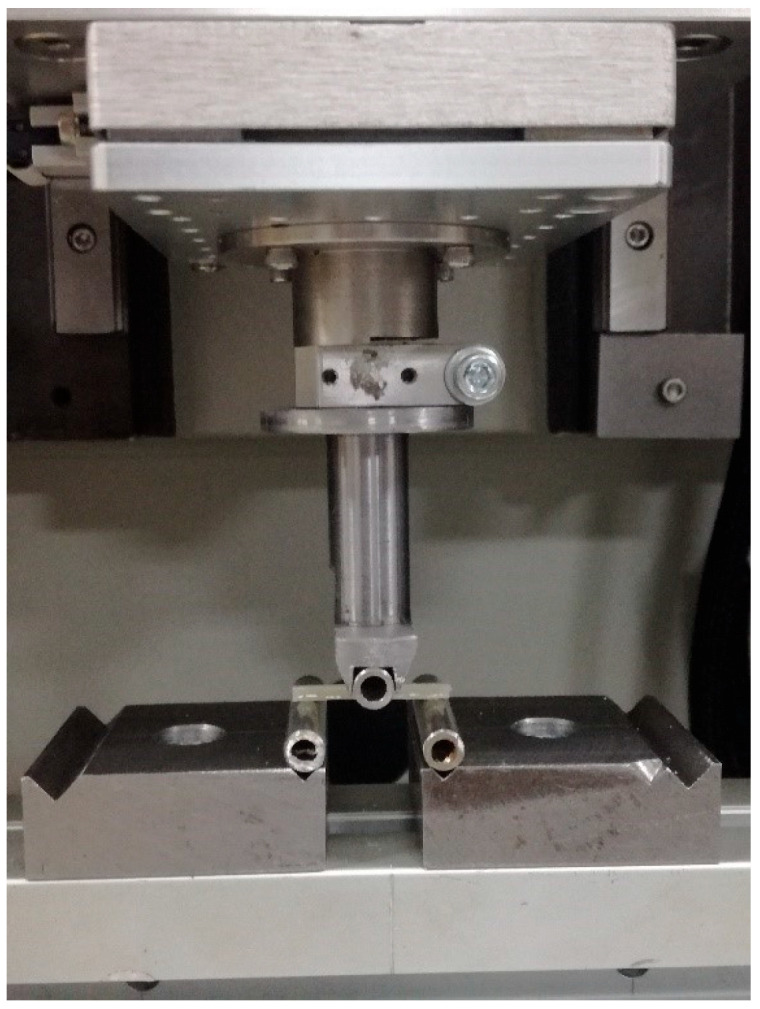
Photo showing the sample placed in the device during the bending test.

### 2.2. Compression Strength Test

For the compression strength test, 10 cylindrical samples were created from each material series with a diameter of 4 mm and a height of 6 mm. This test was carried out on a Walter + Bai testing machine (Walter + Bai AG, Lohningen, Switzerland). The compressive strength was calculated according to the formula:δ = F/πr^2^
where

δ—compressive strength [MPa];

F—destructive force [N];

r—sample radius [mm].

Figure 2 shows a photo of a sample placed in the device during the test.

### 2.3. Diametral Compression Strength Test (DTS)

For the diametrical compressive strength test, 10 cylindrical samples were created from each material series with a diameter of 4 mm and a height of 6 mm. The DTS test was performed on a Zwick/Roell Z020 universal testing machine (Zwick/Roell, Ulm, Germany) at a traverse speed of 1 mm/min. The strength value was calculated according to the following formula:DTS = 2P/πDT
where

P—compressive force that caused the destruction of the material structure and surface [N];

D—sample diameter [mm];

T—sample thickness [mm].

Figure 3 shows a photo of a sample placed in the device during the test.

### 2.4. Impact Strength Test

For impact strength measurements, 10 cuboid samples (5 mm × 10 mm × 20 mm) were made from each material series. The tests were performed using a HIT 5.5p Zwick/Roeler impact hammer (Zwick/Roell, Ulm, Germany) at a hammer energy of 5.5 J. The impact strength of the material was calculated according to the formula:U = A/(b·h)
where

U—the impact strength of the sample [J/m^2^];

A—breaking work (hammer energy) [J];

b—sample width [cm];

h—sample thickness [cm].

Figure 4 shows a photo of a sample placed in the device during the test.

### 2.5. Hardness Measurements

Hardness measurements were carried out on the surface of 10 cuboid samples (10 mm × 20 mm × 5 mm), with 5 measurements taken in randomly selected places. A Shore type D hardness tester (Elcometer Inc., Warren, MI, USA) was used. Shor hardness indicates the resistance of the tested material penetrated by the needle. The value was read on the Shor durometer scale. Figure 5 shows a photo of a sample placed in the device during the test.

### 2.6. Tribological Wear Resistance Test

The wear test was performed using a CSM Instruments Tribometer device (CSM Instruments, Freiburg, Germany) with the Tribox program installed, using the following parameters: friction radius 6.75 mm, speed 0.05 m/s, load 1 N, friction distance 100 m. The test was performed at a temperature of 25 °C in an artificial environment according to Fusayama Mayer (2 dm^3^ of distilled water, 0.8 g NaCl, 0.8 g KCl, 1.59 g CaCl_2_•2H_2_O, 1.56 g NaH_2_PO_4_•2H_2_O, 0.01 g Na_2_S•9H_2_O and 2 g urea) [36]. The test was performed on disc-shaped samples with a diameter of 21 mm and a thickness of 2 mm, in a special Teflon holder. Artificial saliva was then added. The friction counter-sample was a 1/8-inch diameter zirconia ball. The wear of the materials was determined by measuring the linear wear in a friction trace based on the surface roughness measurement using the Hommel Waveline 200 profilometer (ITA, Skórzewo, Poland). The wear of the tested composites was calculated as the volume loss of the material, related to the friction path [37,38]. 

After the friction processes was completed, five abrasion marks were created on each sample and the mean cross-sectional area of the marks was calculated. The volume loss of the material was obtained by multiplying this value by the perimeter of the wear trace. The wear factor was calculated from the following formula:kv = V/(F·L) 
where:

k—material consumption factor [m^3^/(N·m)];

V—volume of material used [m^3^];

F—pressing force [N];

L—total friction path [m].

Figure 6: A sample placed in the device during the test.

The obtained test results were subjected to statistical analysis using Excel (Microsoft Office 2010) and Statistica v. 13. The Shapiro–Wilk test of normality was used to evaluate the distribution of individual parameters. In the case of a non-normal distribution, the Kruskall–Wallis test was then used. In the case of a normal distribution, the equality of variances was assessed using Levene’s test. For equal variances, ANOVA with the Scheffe Post Hoc test was used. The adopted significance level was α = 0.05.

## 3. Results

### 3.1. Bending Strength Test

According to the ANOVA test, a statistically significant difference was demonstrated in the bending strength [MPa] (*p* = 0.0000). The results of the Scheffe post hoc test indicated statistically significant differences between 0% HAp and 2% HAp (*p* = 0.0000), 0% HAp and 5% HAp (*p* = 0.0000) and 0% Hap and 8% HAp (*p* = 0.0000) in all comparisons with larger values in the 0% wt. HAp samples. Moreover, statistically significant difference was demonstrated between 2% HAp and. 8% HAp (*p* = 0.0000) and 5% HAp and 8% HAp (*p* = 0.0000) with smaller values in 8% HAp (Figure 7).

In all cases, the addition of hydroxyapatite caused a decrease in flexural strength. Samples containing 2% and 5% hydroxyapatite filler yielded similar values.

### 3.2. Compression Strength Test

According to the ANOVA test, a statistically significant difference was demonstrated in the compression [MPa] (*p* = 0.0000). The results of the Scheffe post hoc test indicated statistically significant differences between 0% HAp and 5% HAp (*p* = 0.0004) and 0% HAp and 8% HAp with larger values in 0% HAp. Furthermore, a statistically significant difference was demonstrated between 2% HAp and 5% HAp (*p* = 0.0062) and 2% HAp and 8% HAp (*p* = 0.0037) with larger values in 2% HAp (Figure 8).

In all cases, the addition of hydroxyapatite caused a decrease in compressive strength. Samples containing 5% and 8% hydroxyapatite filler yielded similar strength values.

### 3.3. Diametral Compression Strength Test (DTS)

According to the ANOVA test, a statistically significant difference was demonstrated in the DTS (*p*= 0.0000). The results of the Scheffe post hoc test indicated statistically significant differences between 0% HAp and 2% HAp (*p* = 0.0000), 0% HAp and 5% Hap (*p* = 0.0000), and 0% HAp and 8% HAp (*p* = 0.0000), with larger values in 0% HAp (Figure 9).

In all cases, the addition of hydroxyapatite caused a decrease in compressive strength. Samples containing 5% and 8% hydroxyapatite filler yielded similar strength values.

### 3.4. Impact Strength Tests

According to the ANOVA test, no statistically significant difference was found in the impact resistance [J/cm^2^] (*p* = 0.8304) (Figure 10).

Impact tests indicate that the composites containing hydroxyapatite yield slightly lower values compared to the unmodified material.

### 3.5. Hardness Measurements

The Kruskal–Wallis test indicated significant differences in the hardness [ShoreŚ] (*p* = 0.0021). The post hoc test of multiple comparisons of mean ranks for all trials identified statistically significant differences between 0% HAp and 8% HAp (*p* = 0.0005), with larger values in 0% HAp (Figure 11).

Lower hardness was observed with increased hydroxyapatite content.

### 3.6. Impact Strength Tests Tribological Wear Resistance Test 

The Kruskal–Wallis test revealed significant differences in the tribological wear [10^−4^ mm^3^/Nm] (*p* = 0.0074). The post hoc test of multiple comparisons of mean ranks for all trials found significant differences between 5% HAp and 8% HAp (*p* = 0.0467), with larger values in 8% HAp (Figure 12).

Our findings suggest that the addition of 8% filler significantly reduces the wear resistance of the composite material. Composites that contain a larger amount of hydroxyapatite filler obtain a rougher and less even surface, which may reduce their wear rate.

## 4. Discussion

The study hypothesis was confirmed—the use of hydroxyapatite as a filler affects the mechanical properties of dental composites.

### 4.1. Bending Strength Test

In all cases, the addition of hydroxyapatite caused a significant decrease in flexural strength compared to the reference material. However, no significant differences were found between the materials treated with 2% or 5% HAp by weight. The 8% b/w group demonstrated more than double the strength of the untreated reference material; however, it did not achieve a flexural strength of 50 MPa, as specified by the ISO 4049 Dentistry-Polymer-based restorative materials 2009 standard [18] for fillings based on polymers rebuilding hard dental tissues. However, the 2% b/w and 5% b/w HAp had a flexural strength of 60 MPa, and hence are suitable for dental fillings.

### 4.2. Compression Strength

A key property of materials used for fillings is compression strength. Teeth are largely subject to compressive forces while chewing, with a mean force of about 100–150 N [39,40]. The addition of hydroxyapatite caused a decrease in compressive strength from about 230 MPa to about 100 MPa for the higher substitution values; however, no significant difference was observed for 2% b/w HAp. As there are no standards defining the minimum value of compressive strength of this type of material, all treated materials can be considered acceptable, particularly the samples with 2% b/w HAp. Hybrid composites are characterized by high resistance to compressive stresses [41,42]. This is probably due to the higher packing density of the fillers in the polymer matrix.

### 4.3. Diametral Compression Strength

Tensile strength is the basic mechanical test. However, in the case of brittle materials, it is difficult and sometimes even impossible to determine. Therefore, in order to determine the ability of brittle materials for dental fillings to resist tensile stresses that may occur during masticatory processes, a diametrical compression strength test is used [19]. This test can also be performed on small sample sizes, which is important in the case of expensive materials. However, it should be borne in mind that the obtained values are not identical with those obtained in the classic tensile test. In addition, the test provides useful information regarding the tensile strength of materials in FEM modeling of the behavior of teeth with fillings under load. Although there are no guidelines specifying the minimum strength of materials used in fillings for the reconstruction of hard tooth tissues, the minimum DTS value set by the American Dental Association is 24 MPa, as indicated by standard No. 27 “Resin-based fillings” [31]. In each case, the addition of hydroxyapatite reduced the diametrical compressive strength from about 38 MPa for the starting material to about 25–26 MPa for the remaining groups. Significant differences in strength were found between the initial and modified materials. Similar values were published by Okulus and Voelkel [43]. However, there were no differences between the modified groups. All obtained results appear to satisfy standard No. 27 (above).

### 4.4. Impact Strength

In the case of impact tests, no statistically significant differences were found between the groups. The obtained results approximated 0.07 J/cm^2^, which is lower than those observed for flow composites [44]. 

### 4.5. Hardness

Hardness is an important value for materials as it determines their ability to resist deformation. Despite this, it is only an auxiliary parameter used to determine mechanical and operational values; it is difficult to clearly translate its value into other properties. Even so, due to its ease and speed of measurement, it is often used in research. In this case, an increase in HAp was associated with a decrease in the hardness of the tested samples, falling from 82 ShD for commercial samples to 72 ShD (8% b/w) or 78 ShD (2% b/w and 5% b/w). A significant difference was found between unmodified and 8% b/w HAp, but not for the other groups. Despite the observed slight decreases in hardness, the treated materials appear suitable for use in the reconstruction of hard dental tissues.

### 4.6. Wear Resistance Test

A very important property of dental filling materials is their wear resistance. It should be remembered that the chewing process is associated with friction processes and thus wear, which is especially intensified when grinding harder foods. Wear resistance largely determines the lifetime of the restoration, with high values leading to rapid wear of the filling [19,20]. The addition of hydroxyapatite to the tested composite reduces wear resistance. In addition, greater HAp content is associated with higher consumption. While the wear resistance values for 2% b/w and 5% b/w HAp were acceptable, it was significantly lower for 8% b/w.

Our findings do not fully agree with those of other authors. Akhtar et al. suggest that hydroxyapatite may be a promising filler material in dental filling materials [45]. Another study attempted to improve the mechanical properties of glass ionomers with the addition of HAp, with 10% b/w NHA being found to increase the abrasion resistance of Fuji II LC RMGI material [46]. However, excessive amounts of hydroxyaptite may lead to faster wear of the composite [21,22]. Indeed, the addition of hydroxyapatite to composites in an amount above 5% causes the surface to become rougher and heterogeneous, which may significantly reduce wear resistance [4,5]. Hongquan Zhang [47] and Bartoszewicz [48] indicate that the tribological and mechanical properties of a nanocomposite based on acrylic resin are strongly influenced by the morphology of the particles and their size. As such, it is probable that the differences in wear resistance observed between studies result from differences in the morphology of the hydroxyapatite used. It seems that the use of greater than 5% b/w HAp is pointless.

Our findings clearly show that hydroxyapatite affects the strength properties of the hybrid dental composite. It seems reasonable to introduce HAp at 2% b/w and 5% b/w. Despite the reduction in most strength parameters, the obtained values are within the requirements of the relevant standards. Increasing the amount of filler to 8% b/w disqualifies this material for use. 

However, this should not be taken as a final conclusion. It is known that the properties of composite materials largely depend on the size of the filler particles and its surface properties, which allow for better bonding with the matrix. Filler grains of different sizes can mutually fill empty spaces, which can result in composite reinforcement [42]. Hydroxyapatite can be properly fractionated and added to the composite in such a diverse fraction, which should improve its properties. The addition of an additional filler, HAp, to a commercial dental composite resulted in a change in the ratio of the amount of filler to the amount of polymer matrix (matrix), and the overall increase in the total amount of fillers resulted in a relative decrease in the silanizing agent. This influences the quality of the connection of the filler with the matrix: an insufficient amount results in the deterioration of the connection of the polymer matrix and inorganic filler particles, as confirmed previously [49,50,51]. In addition to the amount of hydroxyapatite filler, its morphology can also influence the final result: its fragmentation and irregular surface significantly increases the surface area requiring silanization. Hence, in order to ensure proper silanization when adding HAp, it is necessary to consider its amount and morphology, as well as to increase the amount of pre-adhesive (silanizing) agents modifying the surface, as demonstrated previously [52]. Another solution that may improve the mechanical properties is to introduce additives that block the propagation of composite matrix cracking (e.g., graphene), which should also improve the properties of dental composites [53].

When trying to improve resistance to tribological wear, it must be considered that adding more than 5% HAp to composites causes the surface to become rougher and inhomogeneous, which may significantly reduce wear resistance [54,55].

The development of a dental filling containing hydroxyapatite is also clinically relevant. The current literature indicates that such materials have antibacterial properties [23,24] and such compounds inhibit the formation of secondary caries located under the dental filling, caused by bacterial microleakage.

## 5. Conclusions

Conclusions were reached as follows:The content of hydroxyapatite (30 µm particle size) has a significant impact on the mechanical properties of a dental composite.The mechanical properties of the composite decreased as the amount of hydroxyapatite filler increased.Of the tested combinations, the best tribological properties were obtained by the composite containing 2% wt. hydroxyapatite.Research shows unequivocally that the addition of hydroxyapatite in the amount of up to 5% by weight is legitimate.HAp is an effective treatment for composites when applied at a low concentration. Further research is needed to identify an appropriate size of HAp particles that can be introduced into a composite, to adequately activate the surface and modification its composition.

## Figures and Tables

**Figure 2 materials-16-04548-f002:**
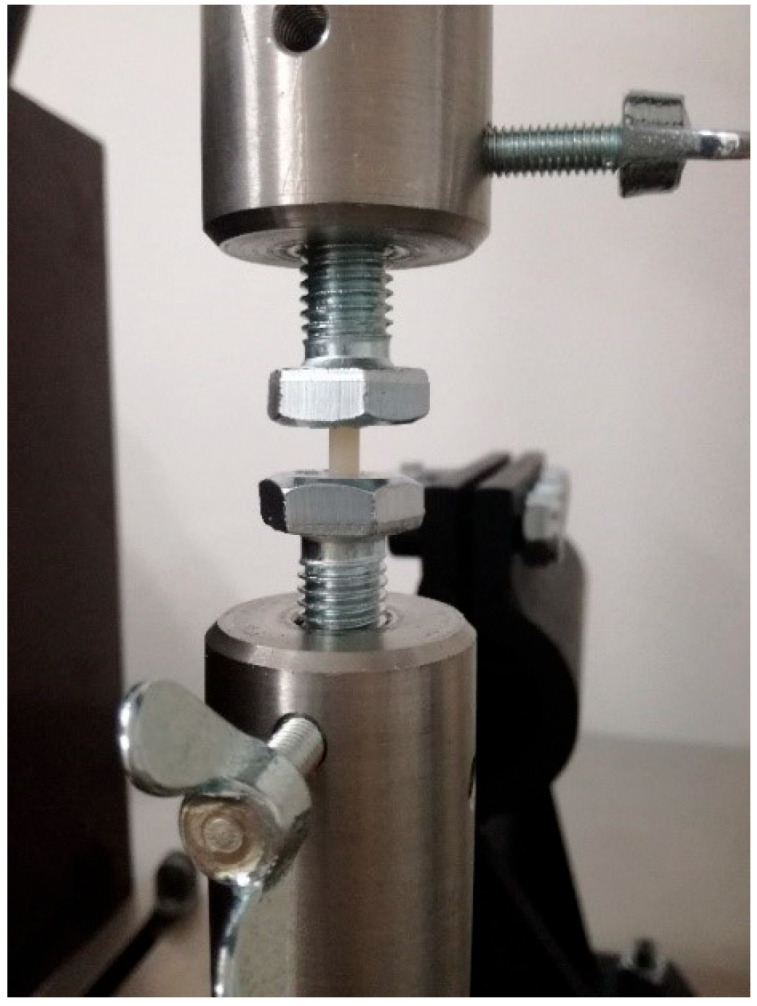
The sample placed in the device during the compression test.

**Figure 3 materials-16-04548-f003:**
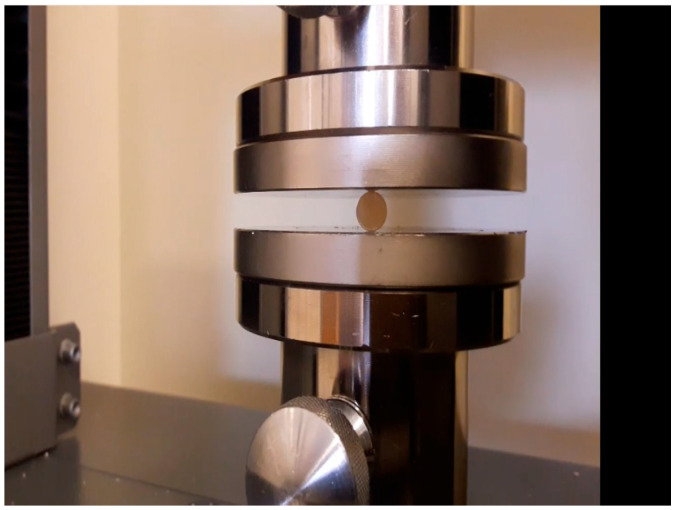
The sample placed in the device during the diametrical compressive strength.

**Figure 4 materials-16-04548-f004:**
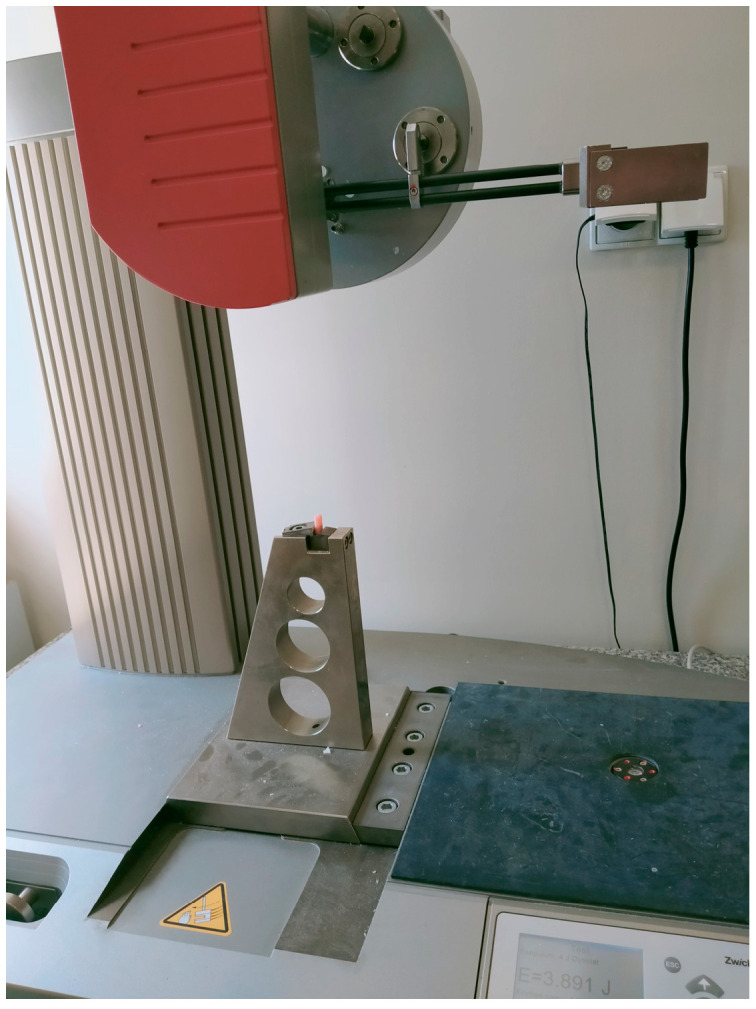
The sample placed in the device during the strength test.

**Figure 5 materials-16-04548-f005:**
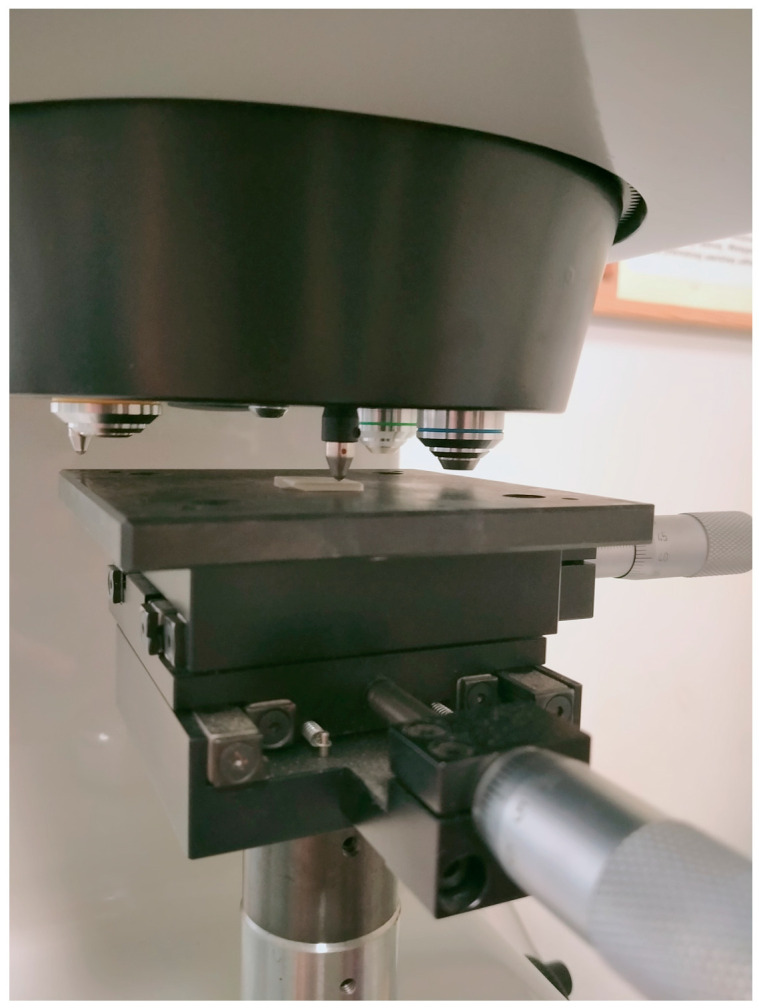
The sample placed in the device during hardness measurements.

**Figure 6 materials-16-04548-f006:**
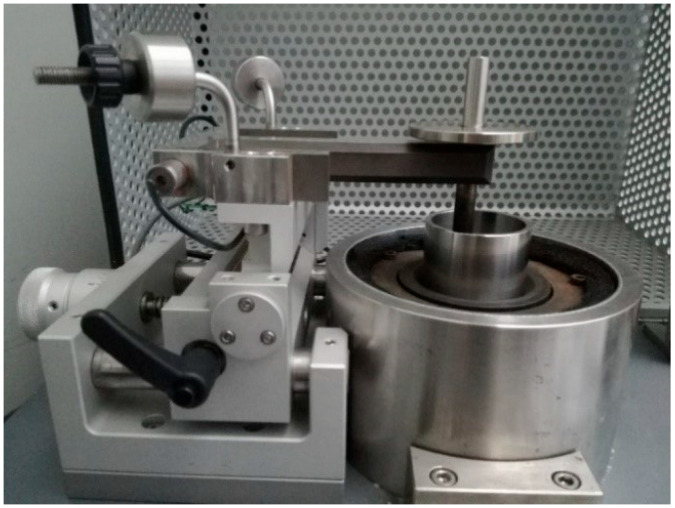
A sample placed in the device during the tribological test.

**Figure 7 materials-16-04548-f007:**
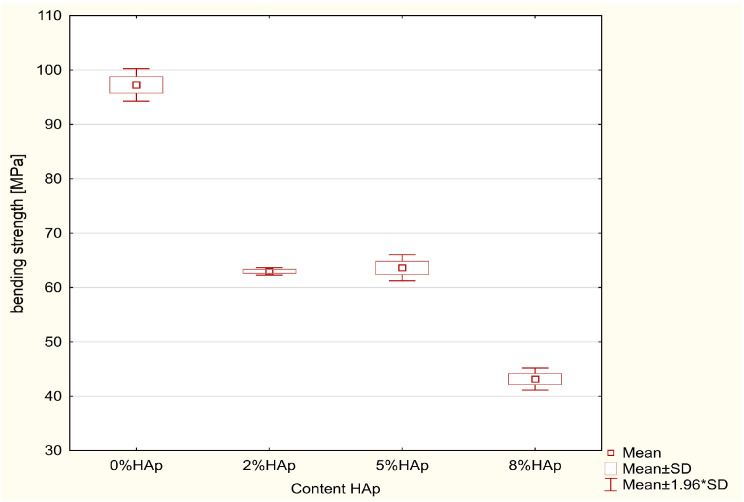
Statistically significant differences between the examined groups for bending strength.

**Figure 8 materials-16-04548-f008:**
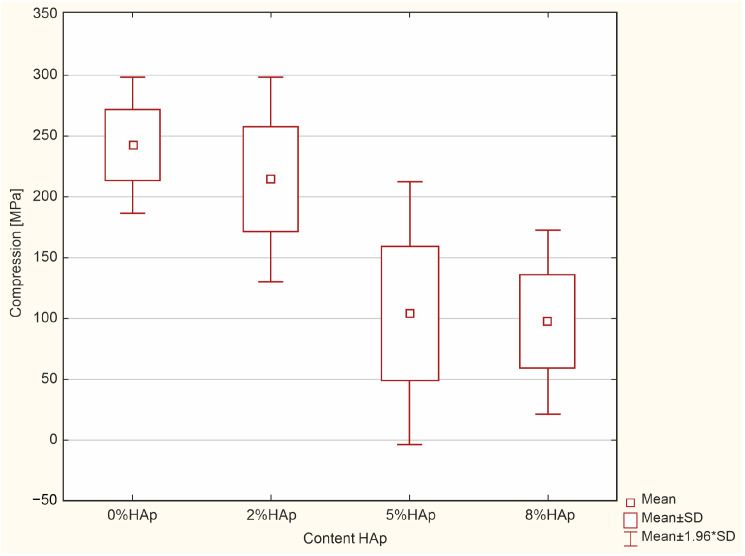
Statistically significant differences between the examined groups for compression.

**Figure 9 materials-16-04548-f009:**
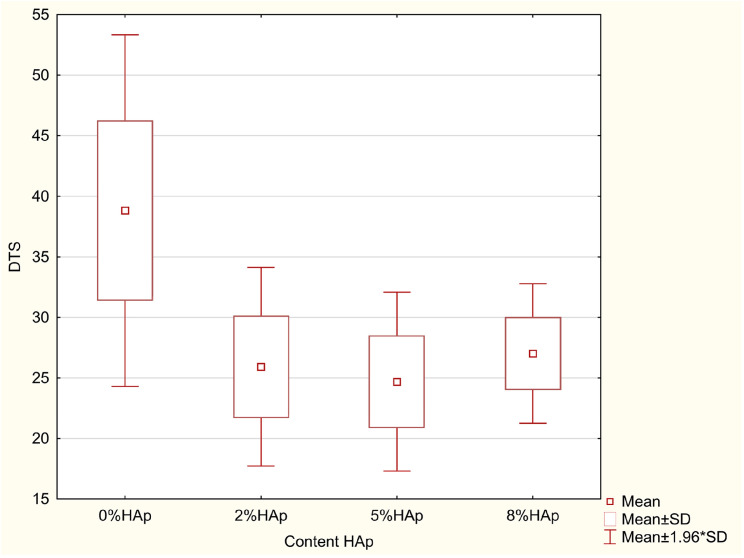
Statistically significant differences between the examined groups for DTS.

**Figure 10 materials-16-04548-f010:**
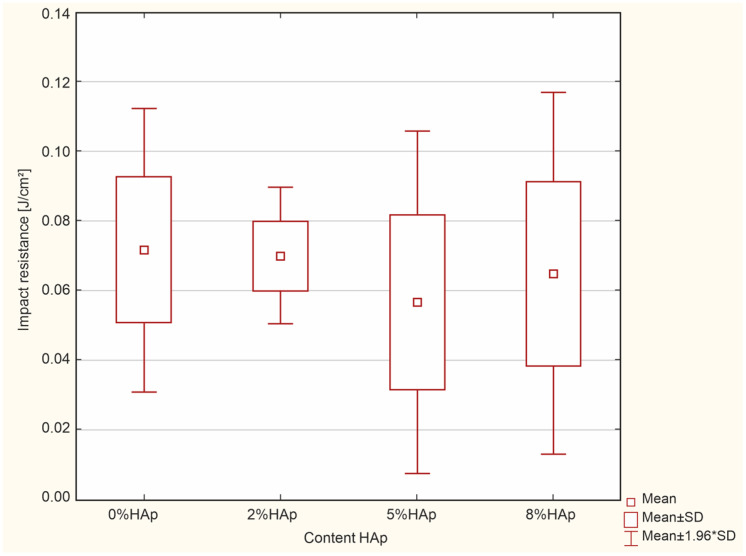
Insignificant differences between the examined groups for impact resistance.

**Figure 11 materials-16-04548-f011:**
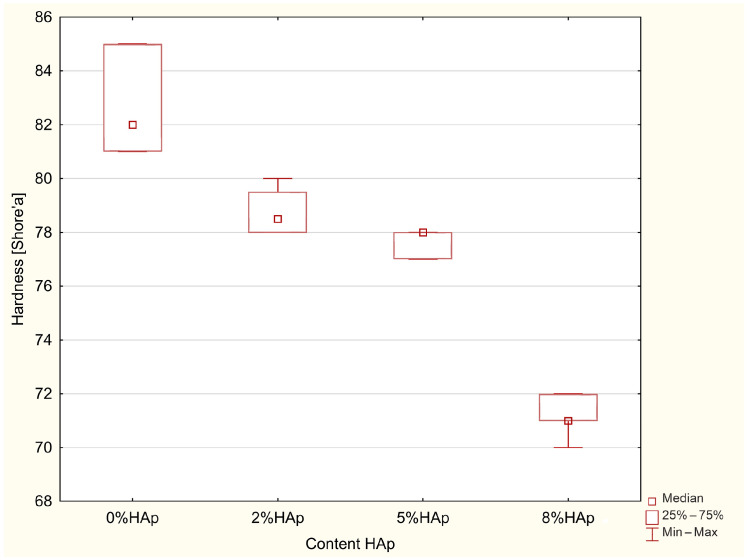
Statistically significant differences in examined groups in hardness.

**Figure 12 materials-16-04548-f012:**
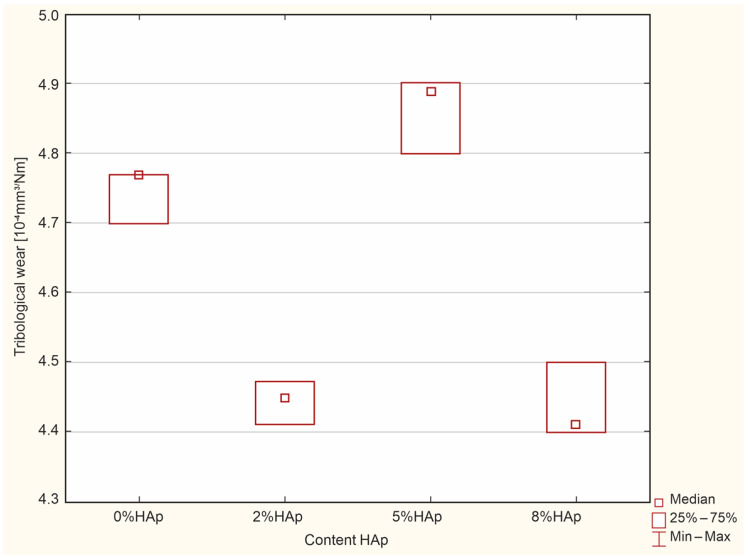
Significant differences between the examined groups for tribological wear.

**Table 1 materials-16-04548-t001:** Filler content and size in individual samples.

Sample Symbol	Composite Type	Resin Type	Filler Content HAp [%] wag.	Filler Size HAp [µm]
HAp 0	light-curing	UDMA	0	-
HAp 2	light-curing	UDMA	2	30
HAp 5	light-curing	UDMA	5	30
HAp 8	light-curing	UDMA	8	30

**Table 2 materials-16-04548-t002:** Test methods, devices, and the shape and size of the samples used in the tests.

Research Method	Devices	Dimensions and Shape of Samples
Bending Strength Test	UMT TriboLab Bruker multifunctional device (Bruker, Karlsruhe, Germany).	Rectangular beam with dimensions of 2 mm × 2 mm × 25 mm
Compression Strength Test	Walter + Bai testing machine (Walter + Bai AG, Lohningen, Switzerland).	A cylinder with a diameter of 4 mm and a height of 6 mm
Diametral Compression Strength Test (DTS)	Universal testing machine (Zwick/Roell, Ulm, Germany)	A disc with a diameter of 4 mm and a thickness of 2 mm
Impact Strength Test	HIT 5.5p Zwick/Roeler impact hammer (Zwick/Roell, Ulm, Germany)	A cuboid with dimensions of 5 mm × 10 mm × 20 mm
Hardness Measurements	Shore type D hardness tester (Elcometer Inc, Warren, MI, USA)	A cuboid with dimensions of 10 mm × 20 mm × 5 mm
Tribological Wear Resistance Test	CSM Instruments Tribometer device (CSM Instruments, Freiburg, Germany) with the Tribox program installed, the Hommel Waveline 200 profilometer (ITA, Skórzewo, Poland).	A disc with a diameter of 21 mm and a thickness of 2

## Data Availability

Not applicable.

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
