# Peer review of "An Evaluation of the Mechanical Properties of a Hybrid Composite Containing Hydroxyapatite"

_materials, 2023, doi:10.3390/ma16134548_

Round 1
Reviewer 1 Report
Journal: Materials (ISSN 1996-1944)
Manuscript ID: materials-2408946
Review Report 1#
The authors presented an article on “An evaluation of the mechanical properties of a hybrid composite containing hydroxyapatite”. The subject of the article falls within the scope of the journal "Materials". However, the article will be ready for publication after a major revision. Comments are listed below.
1. The similarity rate is 31%. It should be reduced.
2. A sentence about numerical results should be added to the abstract part.
3. What's new in this article? How is it different from similar studies in the literature? The original side of the article should be stated in the last paragraph of the introduction.
4. The references given in the introduction are not sufficient. More references should be added.
5. Samples and test chart should be attached in the material and method section.
6. According to which standards were the parameters used in mechanical and wear tests selected?
7. All results were evaluated statistically only. Experimental results should be analyzed in depth and compared with similar studies in the literature and discussed.
8. The data obtained in the Conclusions section should be presented in a more concrete way. Conclusions seem incomplete.
9. The article contains numerous typographic and language errors. It should be corrected.
10. The article should be rearranged by taking into account the journal writing rules and citation rules.
*** Authors must consider them properly before submitting the revised manuscript. A point-by-point reply is required when the revised files are submitted.
Author Response
The authors presented an article on “An evaluation of the mechanical properties of a hybrid composite containing hydroxyapatite”. The subject of the article falls within the scope of the journal "Materials". However, the article will be ready for publication after a major revision. Comments are listed below.
Answer: Thank you very much for your time and comments regarding the article. Your suggestions have added substantive value to the article for the audience.
- The similarity rate is 31%. It should be reduced.
Answer: Most of the similarities concern the separation of materials and methods, we are not able to change the research methodology.
- A sentence about numerical results should be added to the abstract part.
Answer: A sentence about numerical results has been added to the abstract part
- What's new in this article? How is it different from similar studies in the literature? The original side of the article should be stated in the last paragraph of the introduction.
Answer: The original side of the article has been added in the last paragraph of the introduction.
- The references given in the Introduction are not sufficient. More references should be added.
Answer: We added 14 new references in the introduction.
- Samples and test chart should be attached in the material and method section.
Answer: As suggested, two tables have been added to the Materials and Methods section. One of them giving the composition of the samples, and the other giving the research methods and the type of equipment on which the tests were performed. In addition, two photos indicating the devices and samples have been added.
- According to which standards were the parameters used in mechanical and wear tests selected?
- ASTM D2240; Standard Test Method for Rubber Property - Durometer Hardness. ISO: Geneva, Switzerland, 2017.
- All results were evaluated statistically only. Experimental results should be analyzed in depth and compared with similar studies in the literature and discussed.
Answer: The results are analyzed and compared with other similar studies in the Discussion. Unfortunately, as previous works were based on different sample types and methodologies, any comparison between studies is difficult. As suggested, a comment was added to the test results.
- The data obtained in the Conclusions section should be presented in a more concrete way. Conclusions seem incomplete.
Answer: The Conclusions section has been corrected.
- The article contains numerous typographic and language errors. It should be corrected.
Answer: Typographic and language errors have been corrected.
- The article should be rearranged by taking into account the journal writing rules and citation rules.
Answer: The article has been corrected.
*** Authors must consider them properly before submitting the revised manuscript. A point-by-point reply is required when the revised files are submitted.
Reviewer 2 Report
The manuscript is poorly written, too schematic, without specific details, formulas.
The work and the results presentation should be more consistent, with some pictures showing the tested materials. Also these materials should be characterized for other properties, such as roughness, which has a certain influence, as acknowledged by the authors.
Just an example: details about the flexural strength determination are necessary.
Just 6 graphs are not convincing about the work done by the authors.
Author Response
The manuscript is poorly written, too schematic, without specific details, formulas.
Answer: We added 14 new references, specific details and formulas.
The work and the results presentation should be more consistent, with some pictures showing the tested materials. Also these materials should be characterized for other properties, such as roughness, which has a certain influence, as acknowledged by the authors.
Answer: We have added photos showing the tests being performed. Unfortunately, the roughness was not tested.
Just an example: details about the flexural strength determination are necessary.
Answer: We added have information about flexural strength
Just 6 graphs are not convincing about the work done by the authors
Answer: As suggested, two tables have been added to the Material and Methods section. One presents the composition of the samples and the other summarizes the research methods and the testing equipment. Also, photos of the devices and samples have been added.

Reviewer 3 Report
The study is interesting and genuine, however the authors should address the following issues to improve the quality of the manuscript:
- The abstract should be non-structured with specific word limit (please check the authors guidelines).
- The authors should add a short statement in the abstract to emphasize on the current gap in literature and the outstanding research question.
- Null hypothesis/hypotheses should be added at the end of the introduction section.
- Add a table to enlist the information of used materials, machines and softwares (manufacturer, model/reference/LOT numbers...etc).
- Regarding the following statement in line 108: "The radii of the supports and the mandrel implementing the excitation 108 were 1mm", the statement is confusing please edit it for clarity.
- Were sample aged prior to mechanical testing? please rationalize your answer whether it was yes or no.
- The authors should add the clinical significance of the study outcomes at the end of the discussion section.
- The conclusion section should be expanded to cover all the significant outcomes. When possible, please list the conclusion in bullet points.
Author Response
The study is interesting and genuine, however the authors should address the following issues to improve the quality of the manuscript:
- The abstract should be non-structured with specific word limit (please check the authors guidelines).
Answer: The abstract has been corrected.
- The authors should add a short statement in the abstract to emphasize on the current gap in literature and the outstanding research question.
Answer: We have added a short statement in the Abstract to emphasize the current gap in literature and the research question.
- Null hypothesis/ should be added at the end of the introduction section.
Answer: Null hypothesis has been added at the end of the Introduction section.
- Add a table to enlist the information of used materials, machines and softwares (manufacturer, model/reference/LOT numbers...etc).
Answer: The table has been added.
- Regarding the following statement in line 108: "The radii of the supports and the mandrel implementing the excitation 108 were 1mm", the statement is confusing please edit it for clarity.
Answer: We don't understand this comment? In the three-point bending test, both ends of the samples are supported by cylindrical supports with a radius of 1 mm.
- Were sample aged prior to mechanical testing? please rationalize your answer whether it was yes or no.
Answer: No
- The authors should add the clinical significance of the study outcomes at the end of the discussion section.
Answer: The clinical significance has been added.
- The conclusion section should be expanded to cover all the significant outcomes. When possible, please list the conclusion in bullet points.
Answer: This has been corrected as suggested.

Round 2
Reviewer 1 Report
Journal: Materials (ISSN 1996-1944)
Manuscript ID: materials-2408946
Review Report 2#
The authors completed the requested corrections. In my opinion, this article can be accepted for publication in the "Materials" journal in its final form.